# Prenatal Detection of Silver–Russell Syndrome: A First Trimester Suspicion and Diagnostic Approach

**DOI:** 10.3390/medicina61010145

**Published:** 2025-01-16

**Authors:** Slavyana Galeva, Giuliana Diglio, Boris Stoilov, Ekaterina Uchikova, Lucian Pop

**Affiliations:** 1Sheynovo Hospital, 1504 Sofia, Bulgaria; 2Faculty of Medicine, Sofia University, 1504 Sofia, Bulgaria; giuliana.diglio.2070@gmail.com; 3Department of Obstetrics and Gynecology, Medical University of Plovdiv, 4002 Plovdiv, Bulgaria; dr.borisstoilov@gmail.com (B.S.); euchikova@yahoo.com (E.U.); 4Department of Obstetrics and Gynecology, Alessandrescu-Rusescu National Institute for Mother and Child Health, 20382 Bucharest, Romania; popluciangh@icloud.com

**Keywords:** Silver–Russell Syndrome, prenatal diagnosis, asymmetric growth restriction, first-trimester screening, genetic testing, hypomethylation, maternal uniparental disomy

## Abstract

*Background and Objectives:* Silver–Russell Syndrome (SRS) is a rare genetic disorder characterized by prenatal and postnatal growth restriction, distinctive facial features, and body asymmetry. Early suspicion during the first trimester remains challenging but crucial for optimizing clinical outcomes. This study aims to highlight a diagnostic approach to the early suspicion of SRS. *Materials and Methods:* A 28-year-old primigravida presented for routine first-trimester prenatal care. An ultrasound revealed asymmetric growth restriction with normal anatomical findings. The first-trimester biochemical markers, including PAPP-A and β-hCG, were within the normal range. A further evaluation, including amniocentesis and genetic testing, was performed. *Results:* Genetic testing identified hypomethylation at the 11p15 imprinting control region, confirming the diagnosis of SRS. Parental testing excluded the maternal uniparental disomy of chromosome 7, suggesting an epigenetic mechanism. The findings were consistent with a clinical diagnosis of SRS, and appropriate counseling and multidisciplinary management were initiated. *Conclusions:* This case underscores the importance of the early recognition of atypical growth patterns, the integration of advanced genetic testing, and multidisciplinary counseling to guide parental decision-making and improve outcomes.

## 1. Introduction

Silver–Russell Syndrome (SRS), also known as Russell–Silver Syndrome, is a rare genetic disorder characterized by significant prenatal and postnatal growth retardation, distinctive facial features, and body asymmetry. First described in the early 1950s, SRS presents a complex clinical picture that often complicates its diagnosis, particularly in the antenatal setting [1,2]. The syndrome is primarily associated with intrauterine growth restriction (IUGR), where affected infants typically exhibit low birth weight and may present with relative macrocephaly at birth [3,4].

The antenatal diagnosis of SRS is challenging due to the subtlety of its phenotypic manifestations, which may not be fully apparent until later in gestation. However, early suspicion can arise from routine ultrasound examinations that reveal growth discrepancies, such as asymmetric limb development or facial dysmorphisms [5,6]. The Netchine–Harbison clinical scoring system (NH-CSS) has been proposed as a valuable tool for clinical diagnosis, requiring the identification of at least four out of six specific criteria [2,7]. While NH-CSS is primarily designed for postnatal diagnosis, certain criteria—like intrauterine growth restriction (IUGR), relative macrocephaly, and body asymmetry—can raise suspicion during prenatal ultrasound evaluations. NH-CSS remains a postnatal diagnostic tool, its prenatal adaptation relies on growth parameters, biometric ultrasound findings, and genetic investigations to guide clinical suspicion and confirmatory testing.

Genetic testing plays a pivotal role in confirming the diagnosis of SRS, with approximately 60% of cases attributable to epigenetic alterations, particularly hypomethylation at the imprinting control region (ICR) on chromosome 11p15 [3,4,8]. The maternal uniparental disomy of chromosome 7 (UPD(7)mat) is another significant genetic cause, accounting for a smaller subset of cases [3,9]. The integration of genetic testing with clinical assessments can enhance diagnostic accuracy and facilitate appropriate prenatal counseling for affected families.

In this case report, we present a detailed account of a first-trimester suspicion of Silver–Russell Syndrome, highlighting the diagnostic approach and the importance of early recognition in optimizing outcomes for affected infants.

## 2. Case Presentation

We report a case of a 28-year-old primigravida who presented for routine prenatal care with a certain last menstrual period (LMP). Based on her LMP, she was estimated to be at 12 weeks and 4 days of gestation, but the first-trimester ultrasound revealed a crown-rump length (CRL) of 3.91 cm, corresponding to 10 weeks and 6 days of gestation (<3rd percentile). This significant discrepancy between clinical gestational age (CGA) and ultrasound-derived gestational age (UGA) raised concerns about early intrauterine growth restriction (IUGR) with apparent body asymmetry, with the baby’s “head too big for the body”.

### 2.1. First-Trimester Findings

The first-trimester ultrasound scan revealed asymmetric growth restriction, with no overt structural abnormalities detected. The nuchal translucency (NT) measurement was 0.98 mm, and the nasal bone was present (Figure 1).

The first-trimester combined screening test for chromosomal anomalies was performed and analyzed using the LifeCycle 7.0 Software for Prenatal Screening provided by the National Genetics Laboratory, and validated for CRL measurements between 39 and 79 mm. The results were as follows:PAPP-A: 0.7 MoMBeta-hCG: 0.73 MoMNuchal Translucency (NT): 0.98 mm

The calculated risks for chromosomal anomalies were as follows:Risk for Down syndrome (Trisomy 21): 1 in 37,000Risk for Edwards syndrome (Trisomy 18): 1 in 100,000Risk for Patau syndrome (Trisomy 13): 1 in 100,000Risk for Turner syndrome: 1 in 100,000Risk for Triploidy: 1 in 100,000

Initially, there was a suspicion of digynic triploidy due to the significant growth restriction, body asymmetry and gestational age discrepancy. However, the PAPP-A and Beta-hCG levels (0.7 MoM and 0.73 MoM, respectively) did not correspond to the typical biochemical profile observed in digynic triploidy. Furthermore, the absence of major structural defects commonly associated with prevalent chromosomal anomalies in the first trimester increased the suspicion of rarer genetic or epigenetic conditions as the underlying cause of the observed findings.

### 2.2. Why Chorionic Villus Sampling (CVS) Was Not Performed

Despite the suspicion of a genetic disorder, Chorionic Villus Sampling (CVS) was not performed for the following reasons:Epigenetic Nature of SRS: The diagnosis of SRS relies on a DNA methylation analysis at 11p15, which is more accurately assessed using fetal cells from amniotic fluid rather than placental tissue.Risk of Confined Placental Mosaicism (CPM): The placental tissue obtained via CVS might not accurately represent fetal methylation patterns, increasing the risk of false-positive or false-negative results.Diagnostic Accuracy: The amniotic fluid, obtained via amniocentesis, offers greater diagnostic reliability for DNA methylation testing.Patient Counseling: The patient was thoroughly counseled on these factors, and the decision was made to defer invasive testing until amniocentesis could be performed.

### 2.3. Second-Trimester Findings

At 20 weeks of gestation, a follow-up second-trimester ultrasound scan revealed the progression of asymmetric growth restriction, confirmed by the following biometric parameters:Biparietal Diameter (BPD): 42.5 mm (7th percentile)Head Circumference (HC): 162 mm (9th percentile)Abdominal Circumference (AC): 122 mm (<3d percentile)Femur Length (FL): 26 mm (<3d percentile)Estimated Fetal Weight (EFW): 220 g (<3d percentile)Head Circumference-to-Abdominal Circumference Ratio (HC/AC): 1.33

These findings demonstrated severe disproportionate growth restriction, characterized by significant abdominal and femoral length restriction, which were consistent with asymmetric IUGR. No structural abnormalities were identified during the ultrasound examination, and Doppler parameters were within the normal range (Figure 2).

### 2.4. Invasive Genetic Testing

Given the persistent findings of asymmetric growth restriction, an amniocentesis was performed at 20 weeks of gestation. The amniotic fluid analysis included the following:**Chromosomal Microarray Analysis (ChromoSeq)**

ChromoSeq is a comprehensive genetic analysis method used to detect a wide range of genomic abnormalities, including copy number variations (CNVs), structural variants, and single nucleotide changes across the genome. The testing was conducted to rule out the possibility of other rare syndromes. The results of this evaluation were within normal limits.

While ChromoSeq is a powerful tool for detecting a wide range of genomic abnormalities, it is not an effective diagnostic method for Silver–Russell Syndrome (SRS). SRS is predominantly caused by epigenetic abnormalities (e.g., hypomethylation at 11p15.5) rather than classical genetic mutations or chromosomal rearrangements, and these are best detected by specialized techniques such as Methylation-Specific MLPA (MS-MLPA).


**Targeted DNA Methylation Testing for Silver–Russell Syndrome (SRS)**


The genetic testing for Silver–Russell Syndrome (SRS) was conducted using a method called MLPA (Multiplex Ligation-dependent Probe Amplification), specifically the SALSA MLPA KIT ME034-BWS/RSS. MLPA is a molecular genetic technique used to detect copy number variations (CNVs), such as duplications or deletions, as well as DNA methylation changes at specific genetic loci.

SRS is often caused by abnormal DNA methylation at the H19/IGF2 imprinting control region (ICR) located on chromosome 11p15.5.MLPA can quantitatively measure the level of DNA methylation at this specific region.The SALSA MLPA KIT ME034-BWS/RSS targets 11p15 regions associated with both Beckwith–Wiedemann Syndrome (BWS) and Silver–Russell Syndrome (SRS).It assesses methylation levels at H19DMR (differentially methylated region) and KCNQ1OT1DMR.

### 2.5. Testing Process Details


**DNA Extraction:**
○The DNA was extracted from amniotic fluid (prenatal testing) and from parental blood samples.



**MLPA Analysis:**
○Probes specific to the H19DMR and KCNQ1OT1DMR loci on chromosome 11p15 were used.○These probes detect both methylation abnormalities and copy number changes at these regions.


The results confirmed the presence of hypomethylation at the imprinting control region (ICR) on chromosome 11p15, consistent with a diagnosis of Silver–Russell Syndrome (SRS).

### 2.6. Parental Genetic Testing

Both parents underwent genetic testing to exclude the maternal uniparental disomy of chromosome 7 (UPD7mat), a common cause of SRS. Both parents tested negative for UPD, reinforcing that the SRS diagnosis was due to a hypomethylation at chromosome 11p15, which is the most common underlying cause of the syndrome.


**Outcome of the Pregnancy:**


After extensive genetic counseling, the parents opted for the termination of the pregnancy in alignment with national guidelines and their wish.


**Post-Termination Findings:**


A full autopsy was performed following the termination of the pregnancy. The autopsy revealed typical phenotypic features consistent with Silver–Russell Syndrome, including growth restriction, relative macrocephaly, and limb length asymmetry. No major organ defects or structural abnormalities were identified (Figure 3 and Figure 4).

This case highlights the importance of early detection and comprehensive genetic evaluation in cases of suspected Silver–Russell Syndrome, particularly when faced with asymmetric growth restriction in the first trimester. The findings underscore the need for a multidisciplinary approach to prenatal care, enabling timely interventions and informed parental counseling.

This would allow for appropriate counseling to parents and early postnatal specific management, including adequate nutrition and the consideration for growth hormone therapy if the pregnancy is continued.

## 3. Discussion

The case presented highlights the complexities involved in the antenatal diagnosis of Silver–Russell Syndrome (SRS), particularly in the context of first-trimester asymmetric growth restriction. SRS is a genetically heterogeneous disorder characterized by prenatal and postnatal growth failure, relative macrocephaly, and distinctive facial features; however, rare phenotypic manifestations have been described [10,11]. The early identification of growth discrepancies, such as those observed in this case, is crucial for timely intervention and management.

The combination of normal anatomical findings and low-risk results from the first-trimester combined screening test for chromosomal anomalies in our patient is consistent with the findings in the literature, which suggest that first-trimester markers such as PAPP-A and beta-hCG can provide valuable insights into fetal health [12,13]. In this case, the PAPP-A level of 0.7 MoM and beta-hCG of 0.73 MoM indicated a low risk for common aneuploidies, yet the presence of asymmetric growth restriction necessitated further investigation. This aligns with studies indicating that first-trimester growth patterns can be predictive of later fetal growth outcomes [14].

The prenatal diagnosis of first trimester intrauterine growth restriction (IUGR) is challenging and the differential diagnosis is wide. There are few conditions reported in the first trimester associated with fetal body asymmetry and growth restriction [15,16].

Listed below is an overview of the primary differential diagnoses (Table 1).

The decision to perform an amniocentesis was supported by the need for definitive genetic diagnosis, particularly given the suspicion of SRS. The positive results for SRS from targeted testing underscore the importance of genetic evaluation in cases of suspected growth restriction. Previous research has shown that genetic testing can identify the underlying causes in a significant proportion of SRS cases, including a hypomethylation at the imprinting control region on chromosome 11p15 and the maternal uniparental disomy of chromosome 7 [8,12]. In our case, the negative results for UPD in both parents suggest that the SRS diagnosis may be attributed to other genetic mechanisms, which is consistent with the heterogeneous nature of the syndrome. The absence of a maternal uniparental disomy of chromosome 7 (UPD7mat) in this case suggests that the Silver–Russell Syndrome (SRS) diagnosis stems from a hypomethylation at the H19/IGF2 imprinting control region (ICR) on chromosome 11p15.5. This finding indicates a likely de novo epigenetic event, significantly reducing the recurrence risk (<1%) for future pregnancies. Prenatal counseling should emphasize this low recurrence risk while providing guidance on postnatal management strategies, including growth monitoring and potential growth hormone therapy if the pregnancy is continued. Additionally, understanding the absence of UPD7mat refines the clinical focus on epigenetic mechanisms, ensuring accurate diagnosis and targeted care [5,7,8].

Furthermore, understanding the genetic basis of SRS can inform management strategies and potential interventions that may improve outcomes for affected infants. Emerging Next-Generation Sequencing technologies are transforming the field of prenatal diagnostics for Silver–Russell Syndrome. Targeted methylation sequencing and comprehensive whole-genome/exome sequencing can sensitively detect characteristic epigenetic and genetic alterations, such as a hypomethylation at 11p15 and the maternal uniparental disomy of chromosome 7, as well as other rare genetic variants. Despite challenges, including cost considerations and the complexities of data interpretation, NGS holds great promise for enabling early, accurate, and personalized prenatal diagnostics. This technological advancement has the potential to contribute to improved clinical outcomes and facilitate more informed counseling for affected families [17].

A careful consideration of the benefits and limitations of various genetic testing techniques is crucial when selecting appropriate prenatal diagnostic approaches. The choice of test should be based on the specific clinical condition or syndrome suspected, as well as a thorough assessment of the patient’s family and reproductive history [18].

For future pregnancies, the use of multiple diagnostic techniques may be necessary to establish a precise diagnosis. While preimplantation genetic testing holds theoretical potential for detecting genetic and epigenetic abnormalities associated with Silver–Russell syndrome, its clinical application remains limited, and it is not routinely recommended for families without an identified heritable cause [19].

While current non-invasive prenatal testing approaches have limitations in detecting epigenetic alterations associated with Silver–Russell syndrome, such as a hypomethylation at 11p15.5, future developments in methylation-sensitive NIPT techniques may enhance the ability to diagnose this condition prenatally [20].

## 4. Conclusions

This case illustrates the critical role of early detection and comprehensive genetic assessment in the antenatal diagnosis of Silver–Russell Syndrome. The integration of clinical findings, genetic testing, and a thorough understanding of the syndrome’s complexities can enhance diagnostic accuracy and ultimately improve care for affected families.

## Figures and Tables

**Figure 1 medicina-61-00145-f001:**
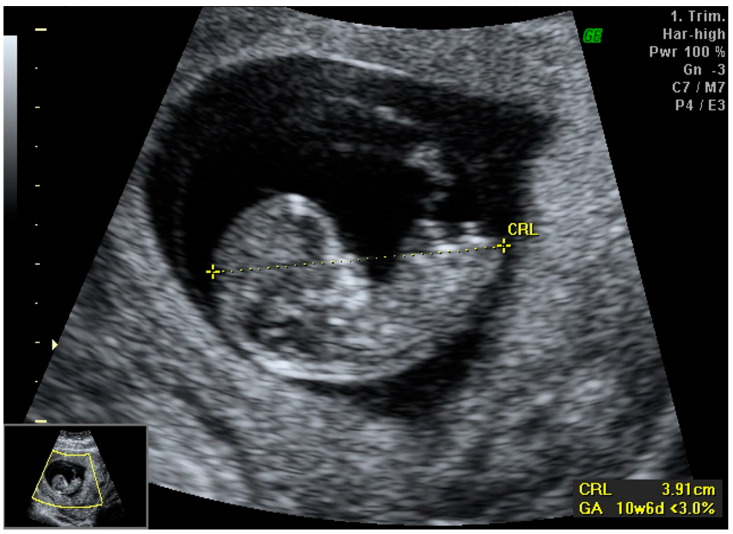
First-trimester scan showing asymmetric fetal growth restriction (CRL: 3.91 cm, corresponding to GA 10 + 6 weeks, <3rd percentile), “head too big for the body”. Imaging conducted with GE Voluson 730 ultrasound system.

**Figure 2 medicina-61-00145-f002:**
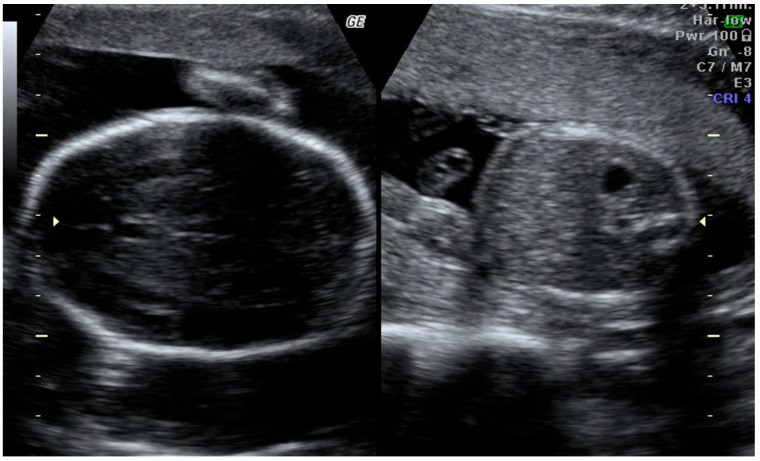
Second trimester scan revealing progression of the asymmetric growth restriction, again “head too big for the body”, the zoom was not altered. Imaging conducted with GE Voluson 730 ultrasound system.

**Figure 3 medicina-61-00145-f003:**
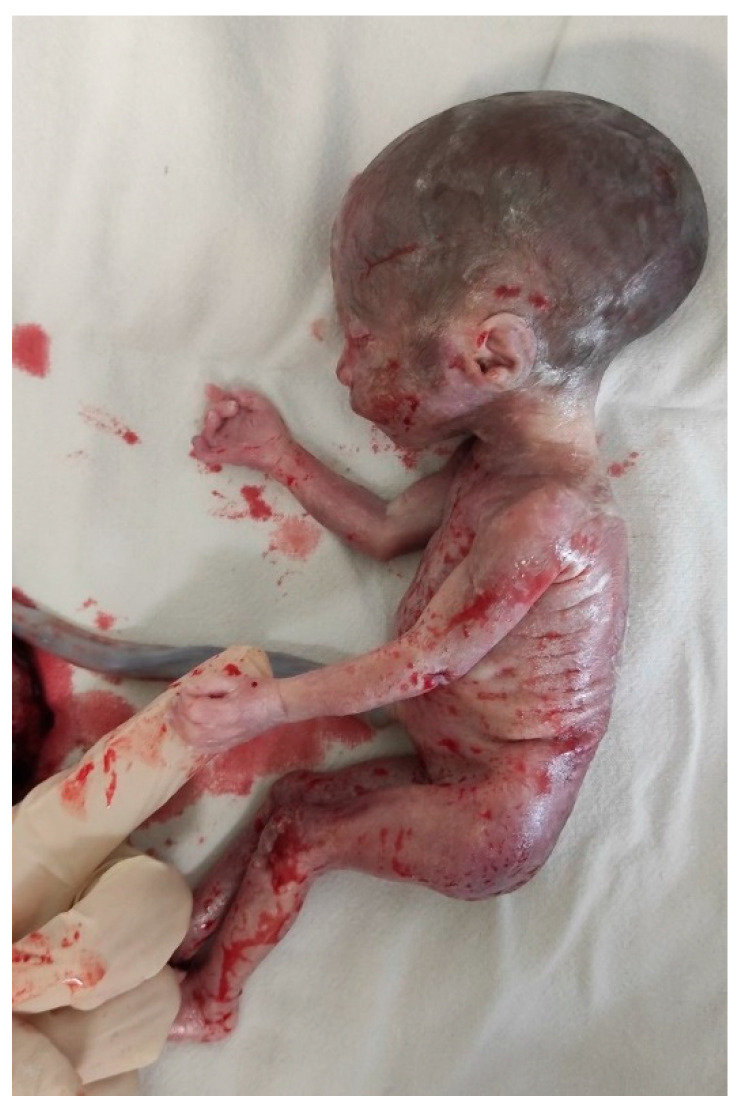
Post-termination findings consistent with ultrasound scan findings—growth restriction, relative macrocephaly, and limb length asymmetry.

**Figure 4 medicina-61-00145-f004:**
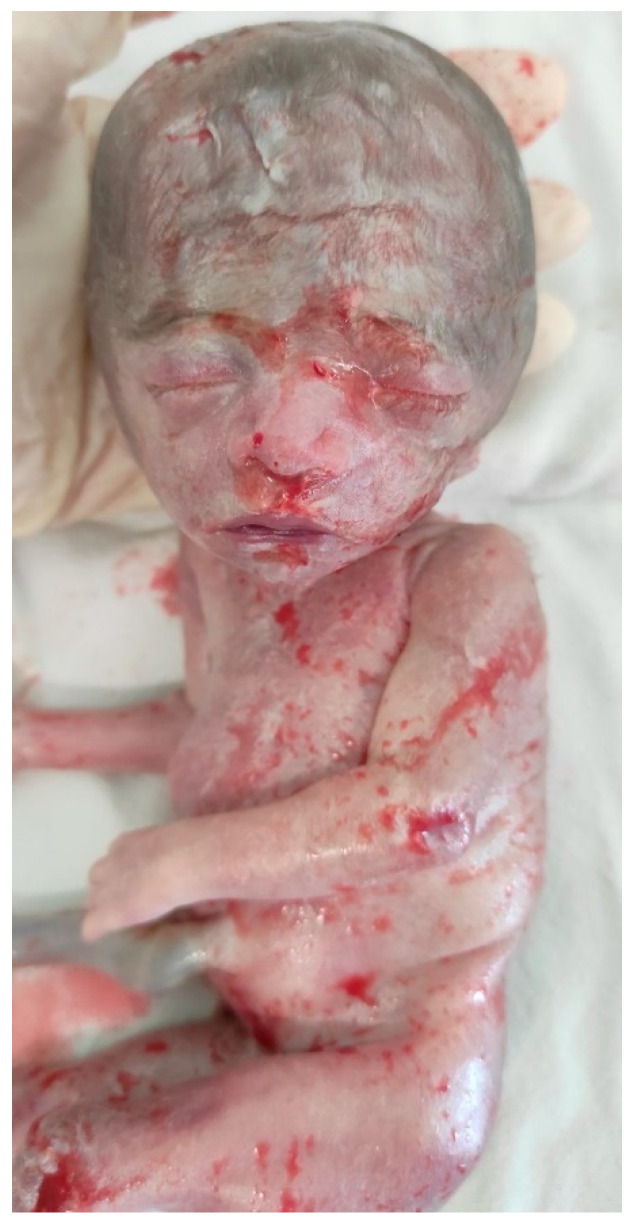
Post-termination findings—facial dysmorphism, triangular face, prominent forehead, and a small jaw.

**Table 1 medicina-61-00145-t001:** Comparison of differential diagnoses for intrauterine growth restriction (IUGR) in the first trimester with biochemical and genetic features.

Condition	Key Ultrasound Findings	Biochemical Markers (PAPP-A, Beta-hCG)	Genetic/Diagnostic Features
Silver–Russell Syndrome	Asymmetric IUGR, relative macrocephaly	PAPP-A: (0.7 MoM), Beta-hCG: (0.73 MoM)—data are limited to our case only	Hypomethylation at 11p15, UPD7 (rare)
Triploidy (Digynic)	Severe IUGR, syndactyly, cystic placenta, holoprosencephaly, exomphalos or posterior fossa cyst	PAPP-A: low, Beta-hCG: low	Triploid karyotype (69,XXX/69,XXY)
Turner Syndrome (45,X)	Mild general growth restriction, cystic hygroma	PAPP-A: low, Beta-hCG: normal	Monosomy X on karyotype
Edwards Syndrome (T18)	Severe general growth restriction, overlapping fingers, cardiac defects, exomphalos, absent nasal bone, single umbilical artery	PAPP-A: low, Beta-hCG: low	Trisomy 18 on karyotype
Patau Syndrome (T13)	Mild growth restriction, megacystis, midline defects, holoprosencephaly, exomphalos	PAPP-A: low, Beta-hCG: low	Trisomy 13 on karyotype
Placental Insufficiency	Symmetric/asymmetric IUGR, oligohydramnios	PAPP-A: low, Beta-hCG: Normal/High	Normal karyotype, abnormal Doppler studies

## Data Availability

The data supporting the findings of this study are available from the corresponding author upon reasonable request.

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
