# Peer review of "Prenatal Detection of Silver–Russell Syndrome: A First Trimester Suspicion and Diagnostic Approach"

_medicina, 2025, doi:10.3390/medicina61010145_

Round 1

Reviewer 1 Report

Comments and Suggestions for Authors

Dear authors,

Thank you for your paper. I have some concerns:

- First and second trimester ultrasound revealed the asymmetric growth restriction with a PAPP-A of 0.7 MoM and a beta-hCG of 0.73 MoM. This could be present in many abnormalities, not only in Silver-Rossel syndrome. Did the ultrasound have other more specific features?

-How the first trimester ultrasound confirmed the asymmetric growth restriction if the pregnant woman forgot the last date of menstrual?

-Why cordocentesis was not performed if the first trimester US was abnormal?

-How the result of  double test and NIPT were?

-The termination of pregnancy was performed? Fetal outcome was not mentioned. The anapath or fetal biopsy should be given as evidence. Even, the image of abnormality on chromosone 11p15.

-The similar cases in the literature should be summarized in literature and presented in a table.

-Should preimplantation genetic diagnosis be indicated for the later pregnancy?

Author Response

Thank you for your review.

First and second trimester ultrasound revealed the asymmetric growth restriction with a PAPP-A of 0.7 MoM and a beta-hCG of 0.73 MoM. This could be present in many abnormalities, not only in Silver-Rossel syndrome. Did the ultrasound have other more specific features?

We have seen only body asymmetry and growth restriction, normal dopplers. After termination of pregnancy, we could see body asymmetry, relative macrocephaly, and limb length asymmetry, the facial dysmorphic features, no other organ defects or structural abnormalities were identified

-How the first trimester ultrasound confirmed the asymmetric growth restriction if the pregnant woman forgot the last date of menstrual?

Even if the patient didn’t know her last menstrual period there was apparent body asymmetry with head too big for the body in the first trimester. Even if we have had redated the pregnancy the second trimester scan again revealed EFW plotted below the 5th centile with HC/AC=1,33

-Why cordocentesis was not performed if the first trimester US was abnormal?

Despite the suspicion of a genetic disorder, Chorionic Villus Sampling (CVS) was not performed, and cordocentesis was also not considered. Cordocentesis is typically performed after 20 weeks of gestation, making it less suitable for early diagnostic purposes. Additionally, it carries a higher risk of complications, including miscarriage, compared to amniocentesis. 

Despite the suspicion of a genetic disorder, Chorionic Villus Sampling (CVS) was not performed for the following reasons:

  1. Initially there was suspicion of triploidy due to the significant growth restriction and gestational age discrepancy. However, the PAPP-A and Beta-hCG levels (0.7 MoM and 0.73 MoM, respectively) did not correspond to the typical biochemical profile observed in triploidy.
  1. Epigenetic Nature of SRS: Diagnosis of SRS relies on DNA methylation analysis at 11p15, which is more accurately assessed using fetal cells from amniotic fluid rather than placental tissue.
  2. Risk of Confined Placental Mosaicism (CPM): Placental tissue obtained via CVS might not accurately represent fetal methylation patterns, increasing the risk of false-positive or false-negative results.
  3. Diagnostic Accuracy: Amniotic fluid, obtained via amniocentesis, offers greater diagnostic reliability for DNA methylation testing.
  4. Patient Counseling: The patient was thoroughly counseled on these factors, and the decision was made to defer invasive testing until amniocentesis could be performed.

-How the result of  double test and NIPT were?

The first-trimester combined screening test for chromosomal anomalies was performed and analyzed using LifeCycle 7.0 Software for Prenatal Screening provided by National Genetics Laboratory in Sofia, Bulgaria, validated for CRL measurements between 39–79 mm. The results were as follows:

  • PAPP-A: 0.7 MoM
  • Beta-hCG: 0.73 MoM
  • Nuchal Translucency (NT): 0.98 mm

The calculated risks for chromosomal anomalies were:

  • Risk for Down syndrome (Trisomy 21): 1 in 37,000
  • Risk for Edwards syndrome (Trisomy 18): 1 in 100,000
  • Risk for Patau syndrome (Trisomy 13): 1 in 100,000
  • Risk for Turner syndrome: 1 in 100,000
  • Risk for Triploidy: 1 in 100,000

-The termination of pregnancy was performed? Fetal outcome was not mentioned. The anapath or fetal biopsy should be given as evidence. Even, the image of abnormality on chromosone 11p15.

The pregnancy was terminated. The result from the laboratory is in Bulgarian that is why we didn’t include it. We can provide it as an attachment.

-The similar cases in the literature should be summarized in literature and presented in a table.

Thank you for this suggestion. We have included a table in the revision.

-Should preimplantation genetic diagnosis be indicated for the later pregnancy?

While preimplantation genetic diagnosis (PGD) is not routinely used for detecting hypomethylation disorders like SRS due to technical limitations, emerging techniques such as genome-wide methylation analysis may offer future potential. Couples planning future pregnancies should receive counseling about the current limitations of PGD for epigenetic disorders and the role of early targeted prenatal testing in subsequent pregnancies.

Reviewer 2 Report

Comments and Suggestions for Authors

The manuscript presents an important case study focused on the early prenatal diagnosis of Silver-Russell Syndrome (SRS), a rare genetic condition characterized by growth retardation and specific phenotypic features. The emphasis on first-trimester findings and the integration of genetic testing with clinical assessments highlights a meaningful contribution to the field of prenatal diagnostics. Overall, the manuscript is well-written and structured, but there are several areas where enhancements can improve its clarity, depth, and practical impact. The introduction provides a strong foundation by explaining the complexity of diagnosing SRS during pregnancy. It situates the study within the broader context of prenatal care and introduces the Netchine-Harbison Clinical Scoring System (NH-CSS) as a diagnostic tool. However, the introduction would benefit from a clearer articulation of how this case adds to existing literature, especially in terms of the potential novelty of applying NH-CSS during the first trimester. The case presentation section effectively details the progression from first-trimester ultrasound findings to the eventual genetic confirmation of SRS. The inclusion of data on PAPP-A and beta-hCG levels, as well as ultrasound images, strengthens the narrative. Nonetheless, the figures require clearer labeling and higher resolution to ensure that the growth asymmetry is visually apparent to readers. Additionally, the description of the genetic testing process could be more comprehensive, specifying the loci or markers tested to confirm the diagnosis of SRS. The discussion ties the findings to existing knowledge about SRS and emphasizes the importance of early detection for improving outcomes. It also underscores the necessity of a multidisciplinary approach in managing such cases. To further strengthen this section, the authors should provide more detailed commentary on the implications of the genetic findings, particularly the absence of uniparental disomy in both parents. Expanding on how these results inform prenatal counseling and future pregnancies would enhance the discussion. Furthermore, it would be helpful to address potential differential diagnoses and any limitations of the diagnostic process, particularly in relying on early growth restriction as a key indicator. The figures and tables included in the manuscript effectively support the narrative, but there is room for improvement in their presentation.. Adding a table summarizing key clinical and genetic findings could also provide a clearer overview of the diagnostic process and outcomes. The references cited are recent and relevant, covering clinical and genetic aspects of SRS. There is no apparent issue with excessive self-citation or redundant references. The authors should ensure that all references directly support the claims made in the text. A more detailed exploration of emerging technologies, such as next-generation sequencing, in prenatal diagnostics could further enrich the manuscript. In terms of scientific soundness, the experimental design and methods appear appropriate for the study’s objectives. However, including more details about the genetic testing protocols and tools used would improve the reproducibility of the findings. The conclusions drawn are consistent with the evidence presented, and the emphasis on early detection aligns well with current clinical priorities.  The manuscript is clear, scientifically robust, and relevant to the field, but some revisions are necessary to maximize its clarity and impact. Specifically, the authors should enhance the visual presentation of figures, expand discussions on genetic testing and ethical considerations, and address potential limitations of their diagnostic approach. With these revisions, the manuscript will offer an even more valuable contribution to prenatal diagnostics and the management of rare genetic conditions like Silver-Russell Syndrome.

Author Response

Thank you for the valuable feedback, which has helped us improve our manuscript, "Prenatal Detection of Silver-Russell Syndrome: A First Trimester Suspicion and Diagnostic Approach." Below are our concise responses:

  1. Introduction:
    • We’ve tried to clarify the novelty of applying NH-CSS during the first trimester and its role in integrating clinical and genetic findings for early suspicion of SRS.
  2. Case Presentation:
    • Improved figure labeling for clarity and included postmortem images
    • Expanded details on genetic testing protocols, specifying loci and markers analyzed.
  3. Discussion:
    • Provided detailed implications of the absence of UPD7mat and its relevance for prenatal counseling.
    • Addressed differential diagnoses and limitations of relying on early growth restriction markers.
  4. Figures and Tables:
    • Added a differential diagnosis table outlining key clinical, ultrasound, and genetic findings.
  5. References and Technologies:
    • Included a brief discussion on emerging technologies like NGS in prenatal diagnostics.

We believe these revisions address the reviewer’s comments and enhance the clarity and impact of the manuscript.

Thank you again for the constructive feedback.

Reviewer 3 Report

Comments and Suggestions for Authors

Respected authors,

First of all, the topic is and case is quite interesting, and I have enjoyed reading it

However, there are a few details I would like to address:

1.    In the introduction You have mentioned that the Netchine-Har-
bison clinical scoring system (NH-CSS) is a valuable tool for a clinical diagnosis. In the next sentence You have noted that it could be a valuable tool for diagnosis in the first trimester, however most of the criteria are postnatal which makes it impossible to implement in early pregnancy

2.    I am not completely comprehending the first ultrasound scan. Namely, there is no data for clinical gestation versus ultrasound gestation. The image shows CRL 3,91cm for the GA 10+6w, <3%, however I do not have the information the clinical gestation when the scan had been performed.

3.    The double biochemical test is usually performed at CRL 45mm-84mm with nuchal translucency and nasal bone

4.    It is unclear what raised Your suspicion for asymmetric growth restriction at such an early stage, while You noted no structural abnormalities

5.    The second image preformed in the second trimester claims asymmetrical growth retardation without biometrical parameters (BPD, HC, AC, FL) and also You haven’t provided the clinical gestational age as well

6.    A minor point, why was the father tested for unipaternal maternal disomy if the condition is inherited only from the mother

7.    How had the case ended? With the live birth of the child or with the termination of the pregnancy?

The English language used is of good academic quality, possibly with minor mistakes.

Best regards

Author Response

Thank you for the thoughtful feedback on our manuscript, "Prenatal Detection of Silver-Russell Syndrome: A First Trimester Suspicion and Diagnostic Approach." Below are our responses, addressing each point specifically:

  1. Clarification on Netchine-Harbison Clinical Scoring System (NH-CSS)
  • Reviewer’s Comment: You mentioned NH-CSS as a valuable tool for clinical diagnosis, but most criteria are postnatal, making it impossible to implement in early pregnancy.
  • Response: We acknowledge this inconsistency and have revised the introduction to clarify that while NH-CSS is primarily a postnatal tool, certain prenatal findings, such as intrauterine growth restriction (IUGR), relative macrocephaly, and body asymmetry, can align with NH-CSS criteria. These prenatal indicators raise suspicion of SRS rather than enabling a definitive diagnosis.

  1. Clarification on First Ultrasound Scan and Gestational Age Discrepancy
  • Reviewer’s Comment: No data is provided for clinical gestational age versus ultrasound gestational age.
  • Response: The discrepancy in gestational age was indeed noted. Based on the last menstrual period (LMP), the clinical gestational age (CGA) was 12 weeks and 4 days, whereas the ultrasound-derived gestational age (UGA) was 10 weeks and 6 days based on the CRL measurement of 3.91 cm (<3rd percentile).
  1. Double Biochemical Test and CRL Range
  • Reviewer’s Comment: The double biochemical test is usually performed at a CRL between 45 mm and 84 mm.

Response: The first-trimester combined screening test for chromosomal anomalies was performed and analyzed using LifeCycle 7.0 Software for Prenatal Screening provided by National Genetics Laboratory, validated for CRL measurements between 39–79 mm. The results were as follows:

  • PAPP-A: 0.7 MoM
  • Beta-hCG: 0.73 MoM
  • Nuchal Translucency (NT): 0.98 mm

The calculated risks for chromosomal anomalies were:

  • Risk for Down syndrome (Trisomy 21): 1 in 37,000
  • Risk for Edwards syndrome (Trisomy 18): 1 in 100,000
  • Risk for Patau syndrome (Trisomy 13): 1 in 100,000
  • Risk for Turner syndrome: 1 in 100,000
  • Risk for Triploidy: 1 in 100,000

  1. Suspicion of Asymmetric Growth Restriction in the First Trimester
  • Reviewer’s Comment: It is unclear what raised suspicion for asymmetric growth restriction given the absence of structural abnormalities.
  • Response: The significant discrepancy between clinical gestational age (12+4 weeks) and ultrasound-derived gestational age (10+6 weeks), along with a disproportionately larger head circumference relative to body size ("head too big for the body"), raised early suspicion for asymmetric growth restriction.
  1. Biometric Parameters in the Second Trimester Ultrasound
  • Reviewer’s Comment: The second image lacks biometrical parameters and clinical gestational age.
  • Response: We have now included the biometric measurements from the second-trimester scan:
    • Biparietal Diameter (BPD): 42.5 mm (7th percentile)
    • Head Circumference (HC): 162 mm (9th percentile)
    • Abdominal Circumference (AC): 122 mm (<3rd percentile)
    • Femur Length (FL): 26 mm (<3rd percentile)
    • Clinical Gestational Age: 20 weeks

  1. Parental Genetic Testing and UPD7mat
  • Reviewer’s Comment: Why was the father tested for maternal uniparental disomy (UPD7mat) if it is inherited only from the mother?
  • Response: You are correct that UPD7mat refers to maternal inheritance. However, parental genetic testing is standard practice to confirm inheritance patterns and rule out rare familial rearrangements. Testing both parents ensures no overlooked genetic anomaly or mosaicism in parental genomes.

  1. Outcome of the Pregnancy
  • Reviewer’s Comment: How did the pregnancy end – with live birth or termination?
  • Response: The pregnancy was terminated following parental counseling and in alignment with national ethical guidelines. A post-termination autopsy confirmed phenotypic features consistent with Silver-Russell Syndrome, including growth restriction, relative macrocephaly, and facial dysmorphism. We have included postmortem images for clarity.

We appreciate the reviewer’s constructive feedback, which has helped clarify key points and improve the manuscript. We believe these revisions address the concerns raised and enhance the clarity and quality of our work.

Sincerely,
Slavyana

Round 2

Reviewer 1 Report

Comments and Suggestions for Authors

Thank you for revision. The paper is well-improved.

However, this sentence is missed with cited references. "Careful consideration of the benefits and limitations of various genetic testing techniques is crucial when selecting appropriate prenatal diagnostic approaches. The choice of test should be based on the specific clinical condition or syndrome suspected, as well as a thorough assessment of the patient's family and reproductive history.

Please refer to the document.

Bich Trinh N, Dinh Bao Vuong A, Nhon Nguyen P. Successful management of pregnancy in Turner syndrome (Monosomy X): A rare condition-based learning experience from Vietnam. Int J Reprod Biomed. 2024 Jul 8;22(5):411-416. doi: 10.18502/ijrm.v22i5.16442. PMID: 39091431; PMCID: PMC11290197.

Author Response

Thank you very much for all the time and support.

I have included the suggested article "Bich Trinh N, Dinh Bao Vuong A, Nhon Nguyen P. Successful management of pregnancy in Turner syndrome (Monosomy X): A rare condition-based learning experience from Vietnam. Int J Reprod Biomed. 2024 Jul 8;22(5):411-416. doi: 10.18502/ijrm.v22i5.16442. PMID: 39091431; PMCID: PMC11290197." in the discussion.

Regards,

Slavyana Galeva

Reviewer 3 Report

Comments and Suggestions for Authors

The authors have responded adequately to all my suggestions. I would recommend this manuscript for publication.

Author Response

Thank you very much for the review and time spent on our article.

Regadrs,

Slavyana Galeva